# Exploring the *Bush yam* (*Dioscorea praehensilis* Benth) as a Source of Agronomic and Quality Trait Genes in White *Guinea yam* (*Dioscorea rotundata* Poir) Breeding

Adeyinka S. Adewumi [1,2], Paterne A. Agre [2,*], Paul A. Asare [1], Michael O. Adu [1], Kingsley J. Taah [1], Jean M. Mondo [2,3] and Selorm Akaba [4]

1 Department of Crop Science, University of Cape Coast, Cape Coast CC-075-8216, Ghana; adewumi.saburi@stu.ucc.edu.gh (A.S.A.); pasare@ucc.edu.gh (P.A.A.); michael.adu@ucc.edu.gh (M.O.A.); ktaah@ucc.edu.gh (K.J.T.)
2 International Institute of Tropical Agriculture, Ibadan 5320, Nigeria; m.mubalama@cgiar.org
3 Department of Crop Production, Université Evangélique en Afrique (UEA), Bukavu 3323, Democratic Republic of Congo
4 Department of Agricultural Economics and Extension, University of Cape Coast, Cape Coast CC-075-8216, Ghana; sakaba@ucc.edu.gh
* Correspondence: P.Agre@cgiar.org

**Abstract:** Yam (*Dioscorea* spp.) is an important food security crop in the tropics and subtropics. However, it is characterized by a narrow genetic base within cultivated and breeding lines for tuber yield, disease resistance, and postharvest traits, which hinders the yam breeding progress. Identification of new sources of desirable genes for these traits from primary and secondary gene pools is essential for this crop improvement. This study aimed at identifying potential sources of genes for yield and quality traits in a panel of 162 accessions of *D. praehensilis*, a semi-domesticated yam species, for improving the major yam species, *D. rotundata*. Significant differences were observed for assessed traits ($p < 0.05$), with *D. praehensilis* genotypes out-performing the best *D. rotundata* landraces for tuber yield (23.47 t ha$^{-1}$), yam mosaic virus (YMV) resistance (AUDPC = 147.45), plant vigour (2.43) and tuber size (2.73). The study revealed significant genotypic (GCV) and phenotypic (PCV) coefficients of variance for tuber yield, YMV severity score, and tuber flesh oxidation. We had also a medium-to-high broad-sense heritability (H$^2$b) for most of the traits except for the dry matter content and tuber flesh oxidation. This study identified some promising *D. praehensilis* genotypes for traits such as high yield potential (WNDpr76, CDpr28, CDPr7, EDpr14, and WNDpr63), resistance to YMV (WNDpr76, CDpr7, EDpr14, CDpr28, and EDpr13), high dry matter content (WNDpr76, CDpr28, and WNDpr24), low tuber flesh oxidation (WNDpr76, CDpr5, WNDpr31, CDpr40, and WNDpr94) and high number of tubers per plant (WNDpr76, CDpr7, CDpr68, CDpr29, and CDpr58). These genotypes could, therefore, be employed in breeding programmes to improve the white Guinea yam by broadening its genetic base.

**Keywords:** wild relative; *D. praehensilis*; *D. rotundata*; inter-specific crosses; yield traits; post-harvest quality; resistance genes

## 1. Introduction

Yam (*Dioscorea* spp.) is an essential food security crop in sub-Saharan Africa, especially in West Africa where more than 95% of the global food yam is produced and ~300 million people depend on its cultivation and trade for food and income [1–4]. In that region, Nigeria, Ghana, Côte d'Ivoire, and Benin are the leading producers with ~50, 8.3, 7.2 and 3.1 million tons, respectively, in 2019 [4]. In these countries, yam provides carbohydrates, proteins, essential minerals, vitamins, and lipids, and it is greatly involved into the social, economic, and religious lives of the local people [5,6]. The annual yam yield in Ghana is ~17.8 t ha$^{-1}$, far below its potential (40–50 t ha$^{-1}$) [4].

Several factors, mostly abiotic (e.g., poor soil fertility, drought stress, etc.) and biotic stresses (e.g., insect pests and diseases such as yam mosaic virus (YMV), yam anthracnose disease (YAD), and yam nematodes) are responsible for the low productivity of cultivated yam species in West Africa [7–11]. Unfortunately, yam is mostly produced by resource-poor farmers who can hardly afford alternative control measures (external farm inputs) such as the use of inorganic fertilizer, pesticides, and irrigation. Developing and deploying improved varieties, combining high yield potential and abiotic and biotic stress resistance, is the most cost-effective and practical way of rising yields in farmer fields in West Africa.

*Dioscorea rotundata*, also referred to as African yam or white Guinea yam, is by far the most cultivated yam species in West Africa. Along with *D. cayenensis* and *D. alata*, they represent more than 95% of produced yam worldwide [1,11]. Farmers and consumers' preferences for a white Guinea yam variety depends on some key traits such as high tuber yield potential, low tuber flesh oxidation/browning, reduced tuber flesh hardening, high dry matter content, and tolerance to YMV and YAD [12]. Accordingly, the genetic improvement of this yam species will have a tremendous impact on food security and poverty alleviation if varieties combining those traits are developed and distributed to the predominantly resource-poor farmers of West Africa. Such effort implies that donor parents for each of the traits are identified within the yam primary and secondary gene pools. Thus, the knowledge of the genetic diversity and ease of gene flow among and within yam species is vital prior designing an inter-specific breeding programme. Based on previous reports, yam breeding has been using a narrow genetic base in developing new varieties for agronomic traits such as resistance to YMV, tuber flesh oxidation and colour, dry matter content, etc., which has resulted in slow progress and low genetic gain in past years [12]. This was partly explained by the vegetative propagation (planting tubers) used for yam cultivation since its domestication. This clonal propagation gradually reduced the genetic diversity and led to the vulnerability to plant diseases and the difficulty of purging deleterious mutations from the germplasms [13]. Broadening the genetic base of existing yam (*Dioscorea* spp.) breeding populations is crucial for increasing the variability and the chance of finding more promising genotypes. Wild relatives are potential sources of resistance, adaptation, and quality trait genes for yam breeding programs and, therefore, a better understanding of their genetic variability is crucial for maximum impact [14,15].

The genus *Dioscorea* consists of ~600 yam species, of which eight are grown in West Africa, where *D. cayenensis* and *D. rotundata* are native and predominant species [16,17]. These two native species emerged from the domestication of wild yams, mainly, *D. praehensilis*, *D. burkilliana*, and *D. abyssinica* [13,18,19]. These wild yam species which are related to the cultivated species, therefore, constitute a vast reservoir of genetic resources that can be exploited to improve the white Guinea yam. In addition, in the era of changing climate, the diversity offered by wild species might provide alternative forms of valuable genes, which could be fundamental in the production of cultivars that are resilient to current and future climatic and edaphic conditions [20]. Wild relatives of cultivated yams might also be the sources of key agronomic and tuber quality traits which can be introgressed as beneficial alleles to improve white Guinea yam and thus broaden its genetic base for breeding in West Africa.

*Bush yam* (*D. praehensilis*) is an edible semi-cultivated wild yam species, utilized primarily for filling the hunger gaps in lean seasons among the local farmers in the forest zones of West African countries such as Nigeria, Ghana, Benin, and Togo [21,22]. This species has a high yield potential, tolerance to insect pests and diseases, longer in-soil storage aptitude, and ability to flower and set fruits profusely [22]. In addition, spontaneous and controlled hybridizations of this species with the white Guinea yam have been reported in West Africa [12,23]. Thus, this species is a promising candidate in the effort of broadening the genetic base of the white Guinea yam and increase in the genetic gain for critical traits.

This study aimed, therefore, at exploring the potential of *D. praehensilis* as a new source for key agronomic and tuber quality traits in white Guinea yam breeding programmes. Specifically, this study sought to: (i) identify *D. praehensilis* genotypes with superior perfor-

mance for nine agronomic and tuber quality traits and (ii) estimate the variance components and broad-sense heritability of those traits.

## 2. Materials and Methods

### 2.1. Experimental Site

The field experiment was conducted between December 2019 and November 2020 at the Teaching and Research farm of the School of Agriculture, University of Cape Coast, Ghana (5°07′7.6″ N, 1°17′18.9″ W, and at 15 m above the sea level). This university is located in Central region of Ghana, a semi-deciduous forest and coastal savannah climatic zones with a bimodal rainfall pattern. The annual rainfall during the study period (December 2019 to November 2020) was 1246.2 mm. The average minimum and maximum temperatures were 24.2 and 28.7 °C, respectively. The average relative humidity for this period was 75.7%. The soil of the experimental site was sandy loam with a slightly acidic pH (6.72), 1.31% organic carbon, 754.6 µg/g available phosphorus, and 0.081 cmol/kg potassium.

### 2.2. Plant Materials

A panel of 174 yam accessions including 162 *D. praehensilis* morphotypes and 12 *D. rotundata* landraces (serving as checks) were used in this study (Table S1). *Dioscorea praehensilis* panel comprised of 72, 24, and 66 morphotypes collected from the Central, Eastern, and Western North regions of Ghana, respectively, while the most preferred *D. rotundata* landraces were sourced from the local markets in Ghana.

### 2.3. Experimental Design and Field Management

The experiment was conducted in simple lattice design with two replicates. The field layout was generated using "Agricolae" package in R software [24]. Each replicate comprised of 18 incomplete blocks with 10 experimental units (plots) as block size. In each replicate, the experimental unit comprised of 3 m long ridges containing three plants at 1 m spacing between and within rows. The planting setts were pre-treated using 70 g Mancozeb (80% WP) as fungicide and 75 mL of cypermetrin (25% EC) as insecticide in 10 L of tap water to prevent soil borne fungi and insect pests from spoiling the setts after planting. The tuber setts from a same accession were labelled properly in net bags and dipped into the solution for 10 min and left in a shaded place for 24 h to allow the cut surface to dry. Hand weeding using hoe was carried out when necessary to reduce the weed competition.

### 2.4. Data Collection

Data were collected on traits of economic significance to yam farmers and consumers. Assessed traits included YMV and YAD severity scores recorded monthly from two to six months after sprout emergence, the plant vigour was assessed four months after sprout emergence, the number of tubers per plot and tuber size were recorded at harvest, tuber yield (per hectare) was collected one year after planting and tuber dry matter content, tuber flesh oxidation and tuber flesh hardness were collected at post harvest. All these traits were assessed using the yam crop ontology recommendations [25].

The plot yield was extrapolated to the yield in tons per hectare using the following formula:

$$TTYH = \frac{TTWP \times 10}{PLS} \tag{1}$$

where TTWP is the total tuber yield per plot and PLS is the plot size.

The dry matter content was determined by chopping 100 g of fresh tuber flesh into small pieces and then oven-dried at 105 °C for 24 h till a constant weight was achieved. The percentage dry matter content was then estimated as:

$$\% \text{ dry matter content} = \frac{\text{Dry tuber flesh weight (g)}}{\text{Wet tuber flesh weight (g)}} \times 100 \tag{2}$$

Intensity of tuber flesh oxidation (colour change or browning of cut tuber flesh) was assessed 60 min after cutting, using a Chroma (colorimeter) meter (CR-400, Konica Minolta, Japan), and the L* (lightness), a* (red/green coordinate), b* (yellow/blue coordinate) values were recorded. A reference of white and black porcelain tiles was used to calibrate the Chroma meter before each reading. The delta (colour difference) (ΔE*) between all the three coordinates was calculated using the following formulas:

$$\Delta E^* = (L^* + a^* + b^*)^{1/2} \tag{3}$$

$$\text{Oxidative browning} = F\Delta E^* - I\Delta E^* \tag{4}$$

where FΔE* is the final delta and IΔE* is the initial delta.

The area under the disease progression curve (AUDPC), a valuable quantitative summary of disease intensity or severity for YMV and YAD over time was estimated using the trapezoidal method [26]. This method discretizes the time variable and calculates the average disease intensity or severity between each pair of adjacent time points:

$$AUDPC = \sum_{i=1}^{N} \left( \frac{y_{i+yi+1}}{2} \right) (t_{i+1} - t_i) \tag{5}$$

where $N$ is the number of observations, $y_i$ is the disease severity at $i^{th}$ observation, $t_i$ is the time at $i^{th}$ observation.

Tuber flesh hardness was assessed with a 6.00 mm probe digital penetrometer. Tuber samples of 1 cm thickness and ~5 cm diameter were prepared from each genotype/accession and the probe was pressed into the tuber. The force necessary for its penetration into the tuber was considered as an indicator of the tuber flesh hardness. Three measurements were taken per accession, the average was calculated, and the data were expressed in Newton. The number of tubers harvested per plant was hand counted and recorded at harvesting.

Data on plant vigour was collected two months after the emergence of the sprouts using the rating scale: 1 = weak (75% of the plants or all the plants in a plot are small, few leaves, and thin vine); 2 = medium (intermediate or normal); 3 = vigorous (75% of the plants or all the plants in a plot are robust with thick vine and leaves very well developed or with abundant foliage).

Data on tuber size was collected at harvest using the rating scale: 1 = small (less than 15 cm length); 2 = medium (between 15 and 25 cm length); 3 = big/large (more than 25 cm length).

*2.5. Statistical Analysis*

A linear mixed model (LMM) for simple lattice design was used to perform the analysis of variance (ANOVA) using *lm* function in R package [27]. The linear model used was as follows:

$$Y_{hijk} = \mu + S_h + G_i + R_j + B_k + \varepsilon_{hijk} \tag{6}$$

where $Y_{hijk}$ = value of the observed quantitative trait in block $k$ and replicate $j$, $\mu$ = population mean; $S_h$ = effect of the $h^{th}$ species, $G_i$ = effect of the $i^{th}$ genotype; $R_j$ = effect of the $j^{th}$ replicate (superblock); $B_k$ = effect of the $k^{th}$ incomplete block within the $j^{th}$ replicate; and $\varepsilon_{ijk}$ = experimental error.

Species and genotypes were considered as random effects while replicates and blocks were considered as fixed effects. Expected mean squares (EMS) from ANOVA using *lmerTest* and *lme4* in R package [27] were employed to estimate the variance components for each trait. Broad-sense heritability ($H^2b$), phenotypic coefficient of variance (PCV), and genotypic coefficient of variance (GCV) were calculated using the values derived from respective variance components. Broad-sense heritability ($H^2b$) was classified as low (<30%), medium (30–60%), and high (>60%), according to Johnson et al. [28]. Based on Deshmukh et al. [29], phenotypic and genotypic coefficients of variation greater than 20%

are regarded as high, whereas values between 10 and 20% are medium and values less than 10% are regarded as low.

The broad-sense heritability (H²b) was estimated using the following formula:

$$H^2b = \left( \frac{\delta_g^2}{\delta_g^2 + \delta_{p/n}^2} \right) \times 100 \tag{7}$$

The phenotypic coefficient of variance (PCV) was determined by:

$$PCV = \frac{\sqrt{\delta_p^2}}{Grand\ mean} \times 100 \tag{8}$$

The genotypic coefficient of variance (GCV) was calculated as follows:

$$GCV = \frac{\sqrt{\delta_g^2}}{Grand\ mean} \times 100 \tag{9}$$

In these formulas, $\delta^2{}_g$ is the genotypic variance, $\delta^2{}_p$ is the phenotypic variance explained as the residual, and the $n$ is the number of replications.

Descriptive statistics, such as the calculations of means, standard deviations, minimum and maximum values and coefficients of variation, were employed to describe variations in key agronomic and tuber quality traits of *D. praehensilis* and *D. rotundata*. Degrees of association among assessed traits were determined using the Pearson's correlation coefficient in R [27]. The association between traits was visualized using corrplot R package version 0.84 [30].

The principal component analysis (PCA) was carried out using the packages Factoextra, and FactoMineR in R [27]. Hierarchical cluster analysis was generated using Pheatmap and Ward.2 methods implemented in Cluster package in R [27]. The silhouette method implemented in Cluster package [27] was used to determine the optimum number of clusters and to assess the effectiveness of grouping. FactoMineR in R package was also used to generate biplots to determine the position of the key agronomic and tuber quality traits of *D. praehensilis* and *D. rotundata* collections. Path coefficient analysis was conducted using FactoMineR in R package considering the tuber yield and tuber dry matter content as response variables. A path diagram was constructed to depict the direct effect of key agronomic and tuber quality traits on tuber yield and dry matter content to determine which traits can be adopted for indirect selection.

## 3. Results

### 3.1. Variability in Key Agronomic and Tuber Quality Traits of D. praehensilis and D. rotundata

Analysis of variance for species revealed significant differences ($p < 0.05$) for most of the parameters evaluated, except the dry matter content, tuber flesh oxidation and the number of tubers per plot (Table 1). Significant differences ($p < 0.05$) were also observed among the genotypes within species for all the traits, except the tuber flesh oxidation (Table 1). *Dioscorea praehensilis* recorded significantly high tuber yield (23.47 t ha⁻¹), low YMV severity score (AUDPC = 147.45), high plant vigour (2.43), and large tuber size (2.73) compared to *D. rotundata* (Table 2). No significant variations were observed in dry matter content and tuber flesh oxidation between the two yam species, although *D. praehensilis* recorded higher maximum values for dry matter content (41.96%) and the number of tubers per plot (~6.00) while *D. rotundata* had better tuber flesh hardness score (39.00) (Table 2).

**Table 1.** Variation due to random effects of agronomic and tuber quality traits in *D. praehensilis* and *D. rotundata* genotypes.

| Sources of Variation | df | Mean Squares | | | | | | | | |
|---|---|---|---|---|---|---|---|---|---|---|
| | | Tuber Yield (t ha$^{-1}$) | DMC (%) | YMV | YAD | TBOXI | TBHard (N) | PLNV | TBRSZ | NTP |
| Replicate | 1 | 946.61 * | 323.41 * | 0.65 NS | $1.28 \times 10^{-24}$ * | 288.63 NS | 0.15 NS | 0.003 NS | 0.003 NS | 7.47 * |
| Block | 14 | 194.37 NS | 9.97 NS | 1704.20 * | 271.42 * | 112.12 NS | 12.21 * | 0.44 * | 0.16 * | 1.28 NS |
| Species | 1 | 1824.46 * | 31.19 NS | 602.71 * | 1814.16 * | 238.16 NS | 2395.04 * | 2.06 * | 1.38 * | 0.69 NS |
| Genotype | 172 | 640.93 * | 19.58 * | 1861.65 * | 553.71 * | 92.37 NS | 2.59 * | 0.54 * | 0.68 * | 1.65 * |
| Residual | 159 | 209.96 | 12.62 | 0.64 | $1.91 \times 10^{-24}$ | 81.98 | 0.37 | 0.01 | 0.01 | 0.74 |

DMC = Dry matter content, YMV = Yam mosaic virus, YAD = Yam anthracnose disease, TBOXI = Tuber flesh oxidation, TBHard = Tuber flesh hardness, PLNV = Plant vigour, TBRSZ= Tuber size, NTP = Number of tubers per plot, SD = Standard deviation, CV = Coefficient of variation. df = degree of freedom, NS = Non-significant, * = Significant at *p*-value < 0.05.

**Table 2.** Mean variations in key agronomic and tuber quality traits of *D. praehensilis* and *D. rotundata* genotypes.

| Species | Tuber Yield (t ha$^{-1}$) ± SD | DMC (%) ± SD | YMV ± SD | YAD ± SD | TBOXI ± SD | TBHard (N) ± SD | PLNV ± SD | TBRSZ ± SD | NTP ± SD |
|---|---|---|---|---|---|---|---|---|---|
| *D. praehensilis* | 23.47 ± 18.53 [a] | 32.83 ± 3.16 [a] | 147.45 ± 31.00 [b] | 267.78 ± 16.48 [a] | −10.36 ± 7.13 [a] | 50.76 ± 1.15 [a] | 2.43 ± 0.52 [a] | 2.73 ± 0.61 [a] | 1.89 ± 0.93 [a] |
| Min. | 1.67 | 21.90 | 135.00 | 210.00 | −35.30 | 39.60 | 1.00 | 1.00 | 1.00 |
| Max. | 123.00 | 41.96 | 270.00 | 315.00 | 4.43 | 53.55 | 3.00 | 3.00 | 5.50 |
| *D. rotundata* | 16.39 ± 10.23 [b] | 34.00 ± 2.82 [a] | 157.50 ± 40.70 [a] | 260.00 ± 23.35 [b] | −6.47 ± 4.25 [a] | 40.10 ± 1.42 [b] | 1.92 ± 0.29 [b] | 2.25 ± 0.45 [b] | 1.71 ± 0.69 [a] |
| Min. | 7.67 | 28.20 | 135.00 | 210.00 | −13.08 | 39.00 | 1.00 | 2.00 | 1.00 |
| Max. | 44.34 | 37.14 | 270.00 | 270.00 | −0.55 | 41.03 | 2.00 | 3.00 | 2.50 |
| CV (%) | 63.40 | 10.79 | 0.53 | 0.27 | 89.67 | 1.22 | 4.94 | 4.35 | 45.90 |

Means followed by the same superscripts are not significantly different using HSD test at *p* < 0.05; SD: Standard deviation. DMC = Dry matter content, YMV = Yam mosaic virus, YAD = Yam anthracnose disease, TBOXI = Tuber flesh oxidation, TBHard = Tuber flesh hardness, PLNV = Plant vigour, TBRSZ= Tuber size, NTP = Number of tubers per plot, SD = Standard deviation.



Coefficients of variation (CV) ranged from 0.27% for YAD severity score to 89.67% for the tuber flesh oxidation. High CVs were recorded for the traits such as tuber yield, tuber flesh oxidation and number of tubers per plot while low CVs were recorded for dry matter content, plant vigour, tuber size, YMV severity, YAD severity, and tuber flesh hardness (Table 2).

### 3.2. Genetic Variability and Broad-Sense Heritability of Agronomic and Tuber Quality Traits in D. praehensilis and D. rotundata

Phenotypic and genotypic variance components, phenotypic and genotypic coefficients of variation and broad-sense heritability of agronomic and tuber quality traits in *D. praehensilis* and *D. rotundata* genotypes are presented in Table 3. Genotypic coefficients of variation (GCV) ranged from 5.8 to 66.3% for tuber flesh hardness and tuber yield, respectively. Phenotypic coefficients of variation ranged from 4.8 to 93.5% for YAD severity and tuber flesh oxidation, respectively. Broad-sense heritability ($H^2b$) varied between 4.9 and 99.9%. High $H^2b$ (>60%) was observed in YMV severity, YAD severity, tuber flesh hardness, plant vigour, and tuber size. Moderate $H^2b$ (30%–60%) were observed in tuber yield and number of tubers per plot, while low $H^2b$ (<30%) were observed in dry matter content and tuber flesh oxidation.

**Table 3.** Genetic variability and broad-sense heritability in *D. praehensilis* and *D. rotundata* accessions.

| | Genetic Parameters | | | | |
|---|---|---|---|---|---|
| Traits | $\delta^2_g$ | $\delta^2_p$ | GCV (%) | PCV (%) | $H^2b$ (%) |
| Tuber yield (t ha$^{-1}$) | 229.6 | 435.4 | 66.3 | 91.3 | 52.7 |
| DMC (%) | 4.0 | 16.6 | 6.1 | 12.4 | 24.1 |
| YMV | 994.8 | 995.4 | 21.4 | 21.4 | 99.9 |
| YAD | 162.0 | 164.0 | 4.8 | 4.8 | 98.8 |
| TBOXI | 4.4 | 89.2 | 20.8 | 93.5 | 4.9 |
| TBHard (N) | 8.5 | 8.9 | 5.8 | 6.0 | 95.5 |
| PLNV | 0.3 | 0.3 | 21.9 | 22.6 | 93.3 |
| TBRSZ | 0.3 | 0.5 | 21.2 | 24.7 | 73.9 |
| NTP | 0.5 | 1.2 | 37.6 | 58.5 | 41.3 |

DMC = Dry matter content, YMV = Yam mosaic virus, YAD = Yam anthracnose disease, TBOXI = Tuber flesh oxidation, TBHard = Tuber flesh hardness, PLNV = Plant vigour, TBRSZ= Tuber size, NTP = Number of tubers per plot, $\delta^2_g$ = Genotypic variance, $\delta^2_p$ = Phenotypic variance, GCV = Genotypic coefficient of variation, PCV = Phenotypic coefficient of variation, $H^2b$ = Broad-sense heritability.

### 3.3. Principal Component Analysis of the Key Agronomic and Tuber Quality Traits

The first three principal components, with eigenvalues greater than one, explained 53.76% of the genotypic variations. The first principal component (PC1) accounted for 23.51% of the total variation and was correlated positively with tuber yield, number of tubers per plot, tuber size, plant vigour, tuber hardness, YAD severity, and dry matter content but it was negatively associated with tuber flesh browning/oxidation and YMV severity (Table 4; Figure 1). The genotypes that contributed positively to the PC1 were: WNDpr76, WNDpr63, CDpr7, EDpr14, CDpr58, WNDpr15, CDpr28, CDpr11, WNDpr79, and EDpr13 (Figure S1). The traits that positively contributed to the second principal component (PC2) were tuber flesh oxidation, dry matter content, and YAD severity, while YMV severity and tuber flesh hardness contributed negatively to PC2 (Figure 1). Accessions such as CDpr50, WNDpr89, Dente, Puna_Central, WNDpr4, Olodo-1, Durban, Dp_Asesewa_UP_E_001, Dp_UP_E_001, WNDpr1, CDpr81, Puna, CDpr23, CDpr1, CDpr54, WNDpr8, CDpr24, WNDpr59, WNDpr41, CDpr75, TDr_Durben, TDr_Mutwu, CDpr10, Dr_Kpanjol, WNDpr56, TDr_Nyaminti, TDr_Asana_North, CDpr85, TDr_Alata_Puna, WNDpr9, WNDpr10, and Dp_Asamankese_Assin_C_002 were positively associated with the PC2 (Figure S1). The variations at the third principal component (PC3) were positively associated with tuber flesh hardness, YAD severity and plant vigour, while YMV severity, tuber yield, and number of tubers per plot had a negative contribution (Table 4).

**Table 4.** Principal component analysis and contributions of agronomic and tuber quality traits on the variability.

| Traits | PC1 | PC2 | PC3 | PC4 | PC5 | PC6 | PC7 | PC8 | PC9 |
|---|---|---|---|---|---|---|---|---|---|
| Yield | 0.0041 | −0.8613 | −0.5015 | 0.0667 | −0.0207 | 0.0171 | −0.0355 | −0.0156 | 0.0015 |
| DMC | −0.0162 | −0.0257 | 0.0094 | 0.1242 | 0.8464 | −0.5164 | −0.0111 | −0.0188 | −0.0059 |
| YMV | 0.9992 | −0.0096 | 0.0268 | 0.023 | 0.0104 | −0.008 | 0.0018 | −0.0002 | 0.0006 |
| YAD | −0.0263 | −0.5051 | 0.8598 | −0.0626 | 0.0018 | 0.0295 | 0.0035 | −0.0026 | −0.0012 |
| TBOXI | −0.0232 | 0.0268 | 0.0886 | 0.9768 | −0.1813 | −0.0611 | 0.0062 | −0.0031 | −0.0084 |
| TBHard | 0.0014 | −0.0205 | 0.0073 | −0.1467 | −0.4997 | −0.8516 | 0.0122 | −0.0353 | −0.0414 |
| PLNV | −0.0011 | −0.0049 | −0.0001 | 0.0027 | −0.0154 | −0.0412 | 0.1198 | 0.0835 | 0.9883 |
| TBRSZ | 0.0001 | −0.0128 | −0.0029 | 0.0009 | −0.003 | −0.0371 | −0.1019 | 0.9913 | −0.073 |
| NTP | −0.0015 | −0.0301 | −0.0219 | −0.0006 | 0.0176 | 0.0068 | 0.9868 | 0.092 | −0.1269 |
| Eigenvalue | 2.115 | 1.516 | 1.207 | 0.994 | 0.876 | 0.765 | 0.712 | 0.557 | 0.258 |
| Variance (%) | 23.505 | 16.843 | 13.412 | 11.040 | 9.736 | 8.504 | 7.913 | 6.184 | 2.864 |
| Cumulative (%) | 23.505 | 40.348 | 53.760 | 64.800 | 74.535 | 83.040 | 90.952 | 97.136 | 100.000 |

DMC = Dry matter content, YMV = Yam mosaic virus, YAD = Yam anthracnose disease, TBOXI = Tuber flesh oxidation, TBHard = Tuber flesh hardness, PLNV = Plant vigour, TBRSZ= Tuber size, NTP = Number of tubers per plot. PC1 to PC9 indicate Principal Components.

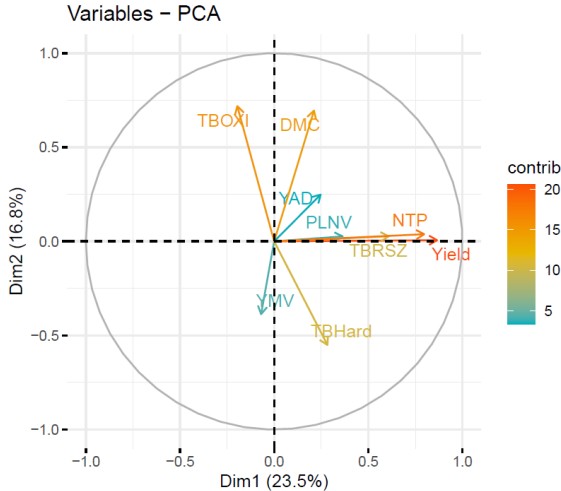

**Figure 1.** Contributions of agronomic and tuber quality traits to PC1 and PC2. DMC = Dry matter content, YMV = Yam mosaic virus, YAD = Yam anthracnose disease, TBOXI = Tuber flesh oxidation, TBHard = Tuber flesh hardness, PLNV = Plant vigour, TBRSZ= Tuber size, NTP = Number of tubers per plot.

*3.4. Relationships among Agronomic and Tuber Quality Traits*

We observed significant correlations among evaluated traits (Figure 2). Tuber yield (t ha$^{-1}$) had significant positive correlations with tuber size (r = 0.38; $p < 0.001$), number of tubers per plot (r = 0.72; $p < 0.001$) and plant vigour (r = 0.16; $p < 0.05$). However, the tuber yield showed significant negative relationship with tuber flesh oxidation (r = −0.13; $p < 0.05$). Tuber yield showed positive but not significant relationship with the dry matter content (r = 0.12) and the YAD severity (r = 0.11). Dry matter content had significant negative correlation with YMV severity (r = −0.16; $p < 0.05$) and tuber flesh hardness (r = −0.17; $p < 0.05$), but showed significant positive correlation (r = 0.23; $p < 0.01$) with tuber flesh oxidation.

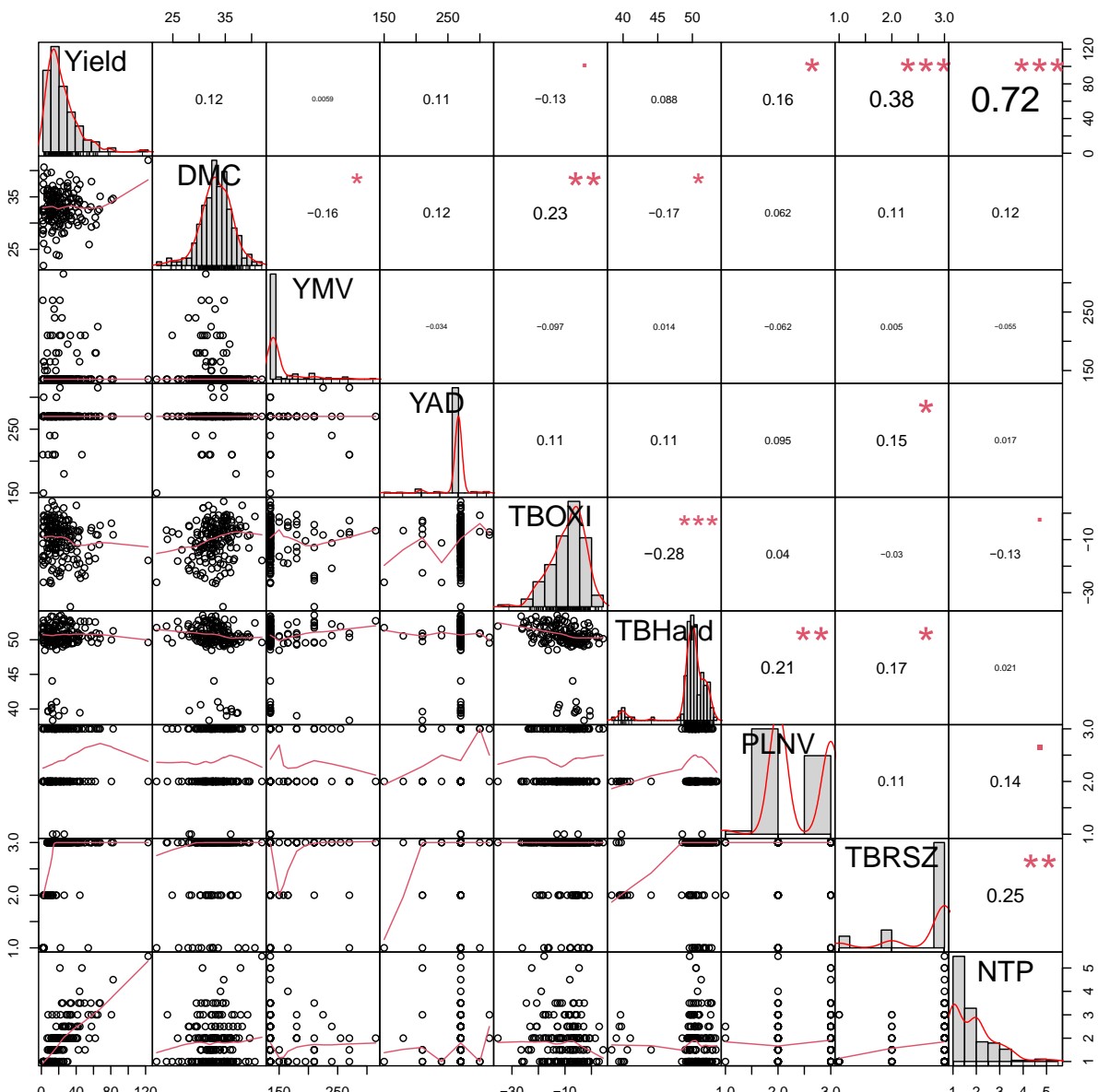

**Figure 2.** Correlation coefficients among agronomic and tuber quality traits. DMC = Dry matter content, YMV = Yam mosaic virus, YAD = Yam anthracnose disease, TBOXI = Tuber flesh oxidation, TBHard = Tuber flesh hardness, PLNV = Plant vigour, TBRSZ= Tuber size, NTP = Number of tubers per plot. Significance level: "$p < 0.1$" = . "$p < 0.05$" = *; "$p < 0.01$" = **; "$p < 0.001$" = ***.

### 3.5. Hierarchical Clustering on Principal Components of D. praehensilis and D. rotundata Genotypes

Hierarchical clustering based on agronomic and quality trait performance grouped the genotypes of *D. praehensilis* and *D. rotundata* into three groups (Figure 3). Cluster 3 had the highest number of genotypes (79), while cluster 1 had the lowest number (39). Hierarchical clustering revealed high significant variation in the distribution of *D. praehensilis* and *D. rotundata* genotypes among the clusters (Table 5). Cluster 1 comprised of genotypes of *D. praehensilis* and *D. rotundata* that possessed low tuber yield (12.19 t ha$^{-1}$), high dry matter content (33.14%), high susceptible to YMV, high YAD resistance, moderate tuber flesh oxidation, minimal tuber flesh hardness and moderate plant vigour, low number of tubers per plot, and small tuber size (Table 5; Figure 3). Among members of cluster 1, we had: Puna, Nyaminti, Puna_Central, Olodo-1, Asana, Dente, Alata_Puna, Nyamint, Durben, Mutwu, and CDpr54 as the genotypes with low tuber flesh hardness; and CDpr50, WNDpr50, WNDpr74, Dente, Puna, WNDpr4, Olodo, WNDpr10, WNDpr9,

and Alata-Puna as the genotypes with high dry matter content. The cluster 2 consisted of
*D. praehensilis* genotypes characterized by high tuber yield (30.91 t ha$^{-1}$), high dry matter
content (33.81%), high resistance to YMV, moderate resistance to YAD, low tuber flesh
oxidation, high tuber flesh hardness, high plant vigour, high tuber size, and high number
of tubers per plot (Table 5; Figure 3). Among cluster 2 members, we had: WNDpr76,
CDpr28, CDpr7, WNDpr63, EDpr14, CDpr58, WNDpr15, CDpr11, and WNDpr79 as supe-
rior genotypes with high yielding ability, WNDpr76, CDpr7, WNDpr63, EDpr14, CDpr58,
CDpr28, CDpr11, WNDpr79, EDpr13, and WNDpr10 as the genotypes with high resis-
tance to YMV, WNDpr76, WNDpr88, CDpr28, CDpr29, WNDpr24, CDpr6, WNDpr84,
CDpr48, WNDpr36, CDpr34, and CDpr5 as the top genotypes with high dry matter content,
WNDpr87, WNDpr36, WNDpr31, WNDpr94, WNDpr21, WNDpr40, WNDpr76, CDpr5,
CDpr6, and WNDpr34 were the top selected genotypes for low tuber flesh oxidation and
WNDpr76, CDpr29, CDpr7, CDpr73, CDpr58, CDpr79, CDpr11, EDpr14, CDpr68, and
EDpr6 were the genotypes with high number of tubers per plot. Cluster 3 contained *D.
praehensilis* genotypes that were characterized by low or no tuber flesh oxidation, high
tuber flesh hardness, high susceptibility to YMV and YAD, moderate tuber yield, large
tuber size, moderate plant vigour, and number of tubers per plot and moderate dry matter
content (Table 5; Figure 3). Of the members of this group, we had: WNDpr68, Cdpr51,
Otim, WNDpr29, EDpr1, CDpr33, WNDpr93, WNDpr7, WNDpr49, and WNDpr19 were
top genotypes with low or no tuber flesh oxidation.

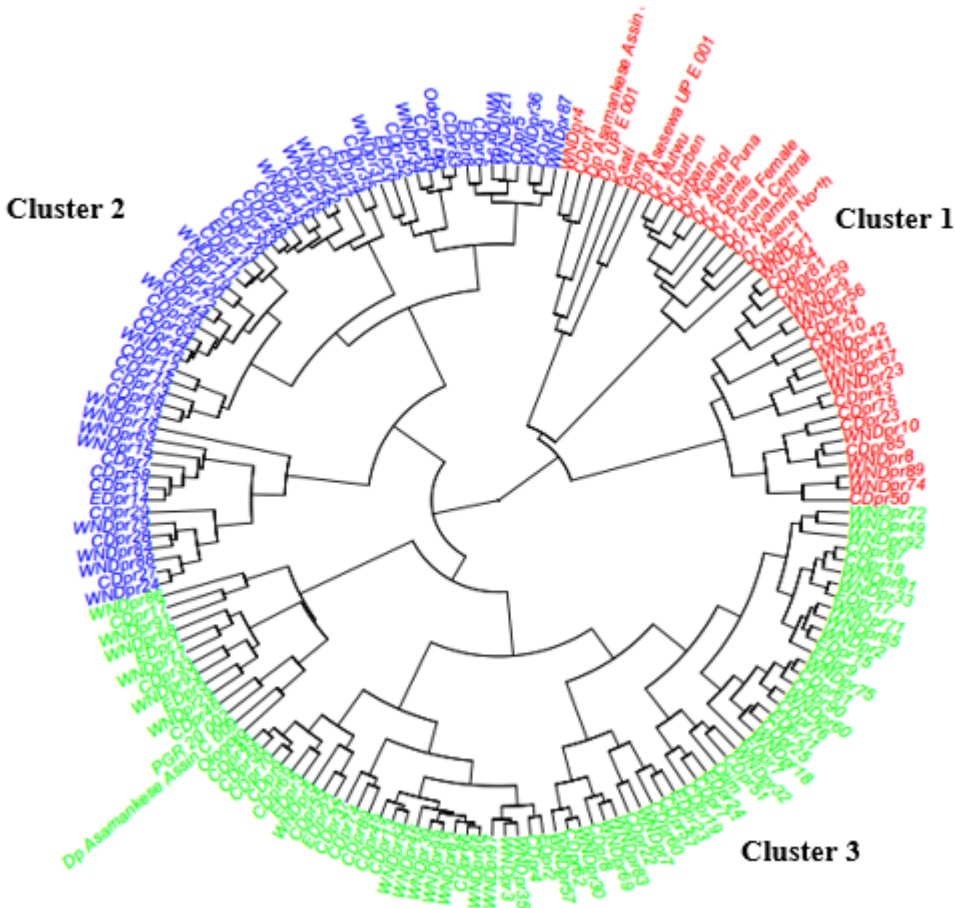

**Figure 3.** Hierarchical dendrogram showing grouping patterns of *D. praehensilis* and *D. rotundata*
genotypes using nine key agronomic and tuber quality traits based on the Gower dissimilarity matrix.
Red, blue and green colours indicate Cluster 1, Cluster 2 and Cluster 3, respectively.

**Table 5.** Description of clusters of *D. praehensilis* and *D. rotundata* genotypes.

| Traits | Cluster 1 ± SD (39) | Cluster 2 ± SD (56) | Cluster 3 ± SD (79) | *F*-value |
|---|---|---|---|---|
| Tuber yield (t ha$^{-1}$) | 12.19 ± 12.70 [c] | **30.91 ± 22.14 [a]** | 22.69 ± 14.20 [b] | 14.11 *** |
| Dry matter content (%) | **33.14 ± 3.67 [ab]** | **33.81 ± 2.60 [a]** | 32.17 ± 3.08 [b] | 4.76 ** |
| Yam mosaic virus | 145.77 ± 31.15 [a] | **137.68 ± 10.36 [b]** | 155.60 ± 39.03 [a] | 5.66 * |
| Yam anthracnose disease | **256.15 ± 31.42 [b]** | 270.54 ± 4.01 [a] | 270.38 ± 8.65 [a] | 11.94 *** |
| Tuber flesh oxidation | −8.17 ± 6.04 [b] | −9.43 ± 6.59 [ab] | **−11.52 ± 7.56 [a]** | 3.42 * |
| Tuber flesh hardness (N) | **47.44 ± 5.07 [b]** | 50.47 ± 0.95 [a] | 50.98 ± 1.31 [a] | 25.30 *** |
| Plant vigour | 2.23 ± 0.48 [b] | **2.96 ± 0.19 [a]** | 2.09 ± 0.36 [c] | 107.15 *** |
| Tuber size | 1.79 ± 0.73 [b] | **2.96 ± 0.19 [a]** | 2.95 ± 0.22 [a] | 133.64 *** |
| Number of tubers per plot | 1.55 ± 0.86 [b] | **2.25 ± 1.03 [a]** | 1.77 ± 0.78 [b] | 8.15 *** |

Significance level: "$p < 0.05$" = *; "$p < 0.01$" = **; "$p < 0.001$" = ***. Means followed by the same superscripts are not significantly different using the least significant difference (LSD) test at 5% *p*-value threshold; SD: Standard deviation. The bold values indicate significant traits at each cluster.

*3.6. Path Coefficient Analysis among Assessed Traits of D. praehensilis and D. rotundata*

The number of tubers per plot and tuber size had highly positive (r = 0.67) and moderately positive (r = 0.21) direct path effects, respectively, on tuber yield per hectare (Figure 4). Dry matter content (r = 0.09), YAD severity (r = 0.08), plant vigour (r = 0.04), and YMV severity (r = 0.03) recorded low but positive direct path effects on tuber yield. The tuber flesh oxidation had low and negative direct path effect (r = −0.03) on tuber yield (Figure 4). In addition, tuber yield (r = 0.24) and tuber flesh oxidation (r = 0.22) had positive moderate path effects on dry matter content, while tuber flesh hardness recorded low negative direct path effect (r = −0.13) on dry matter content (Figure 4).

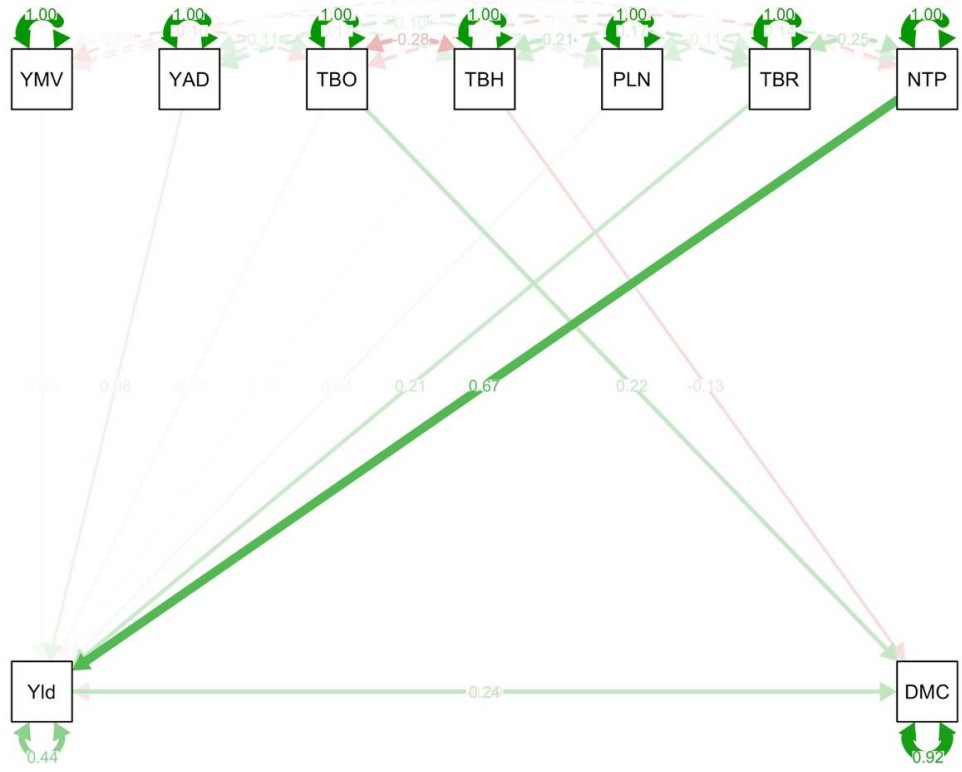

**Figure 4.** Path coefficient analysis between response and independent yam variables. DMC = Dry matter content, YMV = Yam mosaic virus, YAD = Yam anthracnose disease, TBOXI = Tuber flesh oxidation, TBHard = Tuber flesh hardness, PLNV = Plant vigour, TBRSZ= Tuber size, NTP = Number of tubers per plot. Red colour showed negative correlation while the green colour showed positive correlation.

## 4. Discussion

### 4.1. Variability in Key Agronomic and Tuber Quality Traits and Potential of D. praehensilis as Source of Genes for D. rotundata Breeding

White yam production has been constrained by pests and diseases and poor postharvest tuber quality. Unfortunately, breeding populations had shown a narrow genetic base for those traits. Identification of new sources of genes for high yield potential, disease and pest resistance, and good post-harvest tuber quality traits is a prerequisite to the development of varieties that are acceptable by the farmers, consumers, and other end-users. Compared to the cultivated yam varieties, little information is available on the genetic potential of wild yam relatives [31]. The high CVs observed for several traits, including for tuber yield, YMV, tuber flesh oxidation, and the number of tubers per plant, are indicative of the impact of the environment on these traits. Anokye et al. [32] also recorded high CVs for yield traits among Ghanaian water (*D. alata*) yam. Such wide range in trait values and attendant high CVs could serve as a basis for selection in breeding programmes.

The results from this study revealed the existence of a vast genetic variation in the assessed agronomic and tuber quality traits between *D. praehensilis* and *D. rotundata*. High tuber yield observed for *D. praehensilis* compared to *D. rotundata* is an indication that *D. praehensilis* could be used to improve the yield potential of the white Guinea yam. Currently, the yield of the white Guinea yam is ~20% of its attainable yield (40 t ha$^{-1}$) [4,33] and the bush yam could be explored in bridging this yield gap. The high yield of *D. praehensilis* in this study corroborates the findings by [21,22], who reported high yields in *D. praehensilis* after the participatory rural appraisal survey in Togo and Ghana, respectively. Wild yam relatives have also been reported to produce higher yields when compared to the cultivated varieties [31]. The wide range recorded in the agronomic and tuber quality traits are indicative that these traits provide an opportunity for the selection of superior genotypes that can be used for hybridization in yam breeding programmes [34].

From this study, *D. praehensilis* showed more resistance to the YMV when compared with *D. rotundata*. This agreed with the study outcomes of [35], who reported high resistance in *D. praehensilis* in Togo. The high resistance to YMV suggests the existence of resistance genes in the genetic resources of *D. praehensilis* [35] and which can be exploited in *D. rotundata* breeding.

Tuber quality traits are also important traits in the selection and breeding of superior yam varieties [36]. The *D. rotundata* local varieties showed better performance in tuber quality attributes (dry matter content, tuber flesh oxidation, and tuber hardness) than the *D. praehensilis* genotypes; although, some *D. praehensilis* genotypes also recorded comparable tuber quality attributes. The poor tuber quality attributes of *D. praehensilis* have been reported as the major hindrance associated with its disappearance (genetic erosion) from agro systems in Ghana [22]. White yam breeders should, therefore, look for alternative sources of genes for these quality traits.

### 4.2. Genetic Parameters and Broad-Sense Heritability of Assessed Traits

High GCV and PCV (>20%) were observed in some of the evaluated traits such as tuber yield, YMV, tuber flesh oxidation, plant vigour, tuber size, and number of tubers per plant. This is an indication of high selection intensity, which can be imposed on these important traits of the superior genotypes in future yam breeding programmes. High GCV and PCV recorded for tuber yield in this study were in agreement with the findings obtained in the study conducted on advanced breeding population of white yam [37]. Siadjeu et al. [38], also reported high GCV and PCV for harvest index which is a yield component trait in a study conducted on *D. dumetorum* in Cameroon. High H$^2$b (>60%) recorded in this study for traits such as YMV, YAD, tuber flesh hardness, and plant vigour indicates a high correspondence between phenotypic and genotypic variance, and hence, high response to selection. Our results are in agreement with finding of Bhattacharjee et al. [39] and Agre et al. [11] who reported high broad-sense heritability for YAD in water (*D. alata*) yam and YMV in white yam, respectively.

*4.3. Correlation Coefficients, Principal Components, Path Coefficients, and Hierarchical Clusters among Assessed Traits of D. rotundata and D. praehensilis*

Genotypes with high dry matter content, high tuber size, and high number of tubers per plant could be selected for when breeding for improved yield. This was exemplified in the positive correlations between tuber yield and dry matter content, high tuber size and high number of tubers per plot (Figure 2). This corroborates the finding of Agre et al. [40] who reported positive correlation between total tuber weight, tuber shape, and the number of tubers per plant in a panel of water yam. The negative correlation between tuber yield and tuber flesh oxidation suggests that the selection for genotypes with high tuber yield could reduce simultaneously enzymatic oxidation of the tuber flesh. In the present study, no significant correlation was observed between tuber yield and the severity of the two major yam diseases (YMV and YAD). Similarly, from our correlation analysis, the positive correlation that exists between dry matter content and tuber flesh oxidation is an indication that selection for genotypes with high dry matter content will not be affected by increased tuber flesh oxidation. Desirable significant negative correlation observed between dry matter content and YMV severity suggests that the selection of high dry matter content cultivars could reduce the severity of YMV or alternatively, any YMV control measure will help improve yam dry matter content. Weak association of YMV severity with other evaluated traits has also been reported by Asfaw et al. [41] in a study on early generations of breeding population of white yam.

The key agronomic and tuber quality traits that best discriminated the 174 genotypes of *D. praehensilis* and *D. rotundata* were those which resolved on PC1. These traits, including tuber yield, number of tubers per plant, tuber size, plant vigour, tuber hardness, YAD severity, dry matter content, and tuber oxidation could be utilized in evaluating genetic diversity among related *Dioscorea* spp. Agre et al. [12,38,40] has reported the significant contribution of these traits in discriminating yam accessions.

The direct path effects of some of these traits on tuber yield could be utilized for indirect selection in yam breeding programme to enhance the genetic gain in white Guinea yam. Tewedros et al. [42] reported significant direct path coefficients between dry matter content and tuber weight in a study conducted on Ethiopian yam accessions.

The hierarchical clustering in this study revealed similarities among genotypes that were grouped in the same cluster. Clustering of *D. praehensilis* and *D. rotundata* genotypes in cluster 1 (Figure 3) supported the findings of Scarcelli [18] who reported that *D. praehensilis* was the most likely progenitor of white Guinea yam. From our hierarchical clustering, *D. praehensilis* genotypes showed outstanding performance for attributes such as tuber yield, resistance to YMV, tuber size, plant vigour, tuber flesh oxidation, number of tubers per plant, while *D. rotundata* landraces were best for attributes like tuber flesh hardness and resistance to YAD. Development of crosses between promising genotypes of *D. praehensilis* and *D. rotundata* using *D. rotundata* genotypes as female parents could result in development of improved cultivars of white Guinea yam with outstanding performance in important traits like tuber yield, resistance to yam mosaic virus, and some post-harvest tuber quality attributes.

## 5. Conclusions

This study explored 162 accessions of *D. praehensilis* and 12 landraces of *D. rotundata* to identify new sources for key agronomic and tuber quality traits to improve white Guinea yam by broadening its genetic base. We observed wide variations between the two species of yam in terms of tuber yield, dry matter content, resistance to YMV and YAD, tuber flesh hardness, plant vigour, number of tubers per plant, and tuber size. We also observed significant relationships among some traits which can useful for indirect selection. Cluster analysis revealed three groups with contrasting characteristics. This study identified some genotypes of *D. praehensilis* with outstanding performance in tuber yield, resistance to YMV, dry matter content, tuber flesh oxidation, tuber size, number of tubers per plant, and plant vigour. These genotypes could be explored in breeding programmes to improve white

Guinea yam for those traits. Further characterization of this *D. praehensilis* germplasm is required with high throughput molecular markers to refine parental selection prior designing cross-combinations. Combined assessment of these germplasm collection using descriptor keys and molecular markers would provide more insight in the genetic diversity of *D. praehensilis* and its effective use as source of genes to improve white Guinea yam.

**Supplementary Materials:** The following supporting information can be downloaded at: https://www.mdpi.com/article/10.3390/agronomy12010055/s1, Figure S1: PCA-Biplot of key agronomic and tuber quality traits for 174 genotypes of *D. praehensilis* and *D. rotundata*, Table S1: List of *D. praehensilis* and *D. rotundata* genotypes used in the study and their source of collection.

**Author Contributions:** Conceptualization, A.S.A., P.A.A. (Paterne A. Agre), K.J.T. and P.A.A. (Paul A. Asare); methodology, A.S.A.; data analysis, A.S.A. and P.A.A. (Paterne A. Agre); supervision, K.J.T., P.A.A. (Paul A. Asare), P.A.A. (Paterne A. Agre) and M.O.A.; writing—original draft, A.S.A.; manuscript—review and editing, A.S.A., P.A.A. (Paterne A. Agre), P.A.A. (Paul A. Asare), M.O.A., K.J.T., J.M.M. and S.A. All authors have read and agreed to the published version of the manuscript.

**Funding:** We acknowledge funding support from the Bill and Melinda Gates Foundation (BMGF/PP1052998). Also, we acknowledge, funding support from the African trans-regional cooperation through academic mobility (ACADEMY) project, reference number 2017-3052/001-001, funded by the European Union Commission and African Union within the framework of "Intra Africa Mobility Scheme" granted a Ph.D. scholarship to the first author to study at the University of Cape Coast, Ghana.

**Institutional Review Board Statement:** Not applicable.

**Informed Consent Statement:** Not applicable.

**Data Availability Statement:** Data supporting the findings of this work are available within this article.

**Acknowledgments:** Authors acknowledge the provision of research fund to the first author by the ACADEMY project. Directorate of Research, Innovation, and Consultancy, University of Cape Coast is also acknowledged for their contributions.

**Conflicts of Interest:** The authors declare that they have no conflict of interest.

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
