# Peer review of "Exploring the Bush yam (Dioscorea praehensilis Benth) as a Source of Agronomic and Quality Trait Genes in White Guinea yam (Dioscorea rotundata Poir) Breeding"

_agronomy, doi:10.3390/agronomy12010055_

Round 1

Reviewer 1 Report

Overall, this paper is well written and well presented. However, I have some concerns:

- Line 219-220. ggplot is not a package to perform PCA analysis. Rewrite this sentence for clarity.

- Given the total number of accessions (174) used in this study, I was wondering why did you the simple lattice design? How many plants did you used in total? How many times the same accession occurs together in the same block?

- Lines 128-129, You should share the script used to generate field layout with Agricolae” package in R software.

What was the efficient factor?

Line 394-396 You mentioned that your results corroborate findings of other authors but you didn’t in which yam species.

I am surprised that you didn’t mention a pioneer paper on yam assessing Broad sense of heritability, genetic advance, GCV of yam traits including Harvest index  DOI: 10.5897/AJB2014.14067

Author Response

We appreciate the comments from the reviewer we have corrected the manuscript based on the different suggestion 

Reviewer 2 Report

I would recommend to include a reference (introduction, p3, line 82 and the following paragraph) to the recent study (PNAS (2020) 117:31987-31992) which investigates the relationships between D.rotundata and wild relatives D. praehensilis and D. abyssinica by genome re-sequencing. This may have been published after the authors finished working on the present manuscript. 

Author Response

We acknowledged comments from the reviewer and we have provided response to his suggestion in the revised manuscript 
